# GRAPH NEURAL NETWORKS FOR INTERFEROMETER SIMULATIONS

## ABSTRACT

In recent years, graph neural networks (GNNs) have shown tremendous promise in solving problems in high energy physics, materials science, and fluid dynamics. In this work, we introduce a new application for GNNs in the physical sciences: instrumentation design. As a case study, we apply GNNs to simulate models of the Laser Interferometer Gravitational-Wave Observatory (LIGO) and show that they are capable of accurately capturing the complex optical physics at play, while achieving runtimes 815 times faster than state of the art simulation packages. We discuss the unique challenges this problem provides for machine learning models. In addition, we provide a dataset of high-fidelity optical physics simulations for three interferometer topologies, which can be used as a benchmarking suite for future work in this direction.

## 1   INTRODUCTION

Gravitational waves (GWs) are stretches or contractions in the fabric of spacetime, predicted by Albert Einstein in 1916 as a consequence of the general theory of relativity. Gravitational waves enable the study of the most extreme events in our universe, including black hole and neutron star mergers. To that end, multiple ground-based gravitational wave observatories have been built. LIGO is one observatory, which detects GWs using a dual recycled Michelson interferometer (DRMI).

In order to detect GWs, LIGO and future observatories like Cosmic Explorer (CE) require extremely high sensitivities, meaning that the interferometer itself must be robust to real-world errors in manufacturing. Searching for design parameters that are optimally robust is a challenging computational problem, which requires running thousands of costly high-fidelity optical simulations and performing optimization in a high-dimensional, non-convex loss landscape [Richardson et al. (2022)].

Neural networks have been used for data analysis and simulation of complex physical systems in fields from computational fluid dynamics to particle physics to cosmology. However, to date, their use in instrumentation design is less explored. Instrumentation design tasks are usually highly application specific, and have large search spaces and complex design constraints. Furthermore, there are often intricate physical relationships between the design parameters, and simulating a particular set of parameters to evaluate its performance can be extremely computationally expensive.

These challenges make the application of deep learning to instrumentation design tasks particularly appealing. Furthermore, in many cases, the surrogate model does not need to extremely accurately capture the simulation output. For example, in the case of LIGO and CE, it is not necessary for a network to be able to predict the exact field amplitudes with high accuracy; even being able to identify designs that are unlikely to perform well would enable any optimization routine to quickly prune large segments of the parameter space, thus enabling far more efficient design optimization.

We make the following contributions:

1. We demonstrate, for the first time, that deep learning methods can accurately simulate electromagnetic (EM) field propagation in optical cavities at a fraction of the computational cost of traditional methods.

2. We provide a dataset of high-fidelity optical simulations over 3 different interferometer topologies, which can be used for training more advanced models.

## 2 BACKGROUND

### 2.1 GRAPH REPRESENTATION LEARNING

Graph representation learning refers to the task of processing graph structured data with the aim of learning "embeddings" or vector representations for nodes. The graph is encoded as a set of nodes and edges, each of which may have a feature vector. These feature vectors are then aggregated during message passing rounds, resulting in a final feature representation for the node.

In particular, graph attention networks (GATs) [Veličković et al. (2017)] are among the state-of-the-art methods for representation learning. In a GAT, a learned attention module is used to weight messages received from each neighbor in a node's $k$-hop neighborhood, allowing the network to learn to prioritize connections. In particular, attention coefficients are computed using the following formula,

$$\alpha_{ij} = \frac{\exp\left(\text{LeakyReLU}\left(\vec{a}^T\left[\mathbf{W}\vec{h_i}||\mathbf{W}\vec{h_j}\right]\right)\right)}{\sum_k \exp\left(\text{LeakyReLU}\left(\vec{a}^T\left[\mathbf{W}\vec{h_i}||\mathbf{W}\vec{h_k}\right]\right)\right)} \tag{1}$$

where $\vec{h}$ represents the feature vector, $\vec{a}$ is a learnable weight vector, and $\mathbf{W}$ is a learnable weight matrix.

### 2.2 KOLMOGOROV-ARNOLD NETWORKS

Kolmogorov-Arnold networks are an alternative to the multilayer perceptron (MLP), introduced by Liu et al. (2024), that use the Kolmogorov-Arnold Representation theorem to represent multivariate functions as a sum of univariate functions.

$$f(x_1, x_2, ..., x_n) = \sum_{q}^{2n+1} \Phi_q\left(\sum_{p=1}^{n} \phi_{p,q}(x_p)\right) \tag{2}$$

In contrast to MLPs, where the goal is to learn the linear transformations for each layer, in KANs, the goal is to learn the univariate "activation functions," $\phi_{p,q}$. In particular, we employ the FastKAN implementation from Li (2024), which uses radial basis functions to parametrize the functions, $\phi_{p,q}$.

### 2.3 INTERFEROMETER PHYSICS

The LIGO interferometer is comprised of two main arm cavities, in which laser light reflects hundreds of times, before being reflected back to be measured. In nominal operation, the two arms are set up such that the reflected light from each arm should destructively interfere with each other, resulting in no signal being read out. However, when a gravitational wave passes, the length of the arms is stretched or contracted, resulting in imperfect cancellation of the fields at the readout port.

The field at every point in the LIGO interferometer can be decomposed into a linear combination of orthogonal basis functions, the complex-valued Hermite-Gauss (HG) modes

$$E(x, y, z) = u_l(x, z)u_m(y, z)e^{-ikz} \tag{3}$$

where $u_l$ and $u_m$ are

$$u_J(x, z) = \left(\frac{\sqrt{2/\pi}}{2^J J! w_0}\right)^{1/2} \left(\frac{q_0}{q(z)}\right)^{1/2} \left(-\frac{q^*(z)}{q(z)}\right)^{J/2} H_J\left(\frac{\sqrt{2}x}{w(z)}\right) \exp\left(\frac{-ikx^2}{2q(z)}\right) \tag{4}$$

Here, $q$ denotes the complex beam parameter, $w$ denotes the beam waist, and $H_J$ denotes the $J$th Hermite polynomial.

Alternatively, assuming cylindrical symmetry, the field can be decomposed into a linear combination of the Laguerre-Gauss (LG) modes (Bond et al. (2016)), indexed by $p, l$

**a) Fabry Perot**

Laser

Cavity

**b) Simple Coupled Cavity**

Laser

Cavity

**c) Arm-SRC Coupled Cavity**

Laser

Arm Cavity

Signal Recycling
Cavity

**d) DRFPMI**

Laser     Power Recycling
Cavity

Arm Cavity

Signal Recycling
Cavity

Figure 1: The interferometer topologies that we consider in this paper, in order of increasing complexity. a) A Fabry-Perot resonator is the simplest optical cavity. It consists of two curved mirrors, which reflect the light back and forth, building up power inside the cavity. b) A coupled cavity system. Two more mirrors are placed after the Fabry-Perot cavity, creating a second cavity, which must be mode matched to the first. c) a Arm-SRC coupled cavity (CC). This topology contains an a Fabry-Perot resonator, labelled the "arm cavity", and a signal recycling cavity (SRC), which is used to amplify the power in the signal light. d) A dual recycled Fabry-Perot Michelson interferometer. This is the style of interferometer that LIGO is. In addition to the Arm-SRC coupled cavity case, we add the second interferometer arm, and the power recycling cavity (PRC), whose design closely mimics the SRC.

$$u_{p,l}(r, \phi, z) = \frac{1}{w(z)} \left( \frac{\sqrt{2}r}{w(z)} \right)^{|l|} \sqrt{\frac{2p!}{\pi(|l| + p)!}} L_p^l \left( \frac{2r^2}{w^2(z)} \right)$$

$$\exp\left( -ik\frac{r^2}{2q(z)} + il\phi \right) \exp(i(2p + |l| + 1)\psi(z)) \quad (5)$$

where $L_p^l$ denotes the $p, l$th associated Laguerre polynomial, and $\psi(z)$ is the Gouy phase. The power contained in the field at any given plane along the direction of propagation is given by the integrated magnitude of the field,

$$P(z) = \int \int E^*(x, y, z)E(x, y, z)dxdy \quad (6)$$

Instead of directly predicting the power at each point in the interferometer, we can also formulate the problem in terms of optical gain factors:

$$P_{out} = g_{mn}P_{in} \quad (7)$$

where $g_{mn}$ is the optical gain for the $m, n$th mode. The laser produces a pure "Gaussian" beam, meaning that only the $p = 0, l = 0$ HG mode is present. However, through mode mismatches and other scattering sources in the interferometer, power is lost to higher order modes.

Traditionally, this loss is computed by creating a mode scattering matrix for each optical component, and then computing how the incoming field is transformed by the scattering matrix. However, for complex interferometers, with many optical components, this can be extremely computationally expensive.

# 3  RELATED WORK

Classical methods for interferometer simulation rely on rely on representing the optical field in the interferometer, either using a modal decomposition (Bond et al. (2016)) or directly propagating the field.

In the modal decomposition approach, the EM field is represented in a particular basis. In the simulations used to produce the dataset included with this paper, the field is represented in the HG basis of one of the cavities of the particular interferometer. The field propagation is then modelled via a series of "ABCD" matrices, which model how the field amplitude and phase change either as the field passes through free space, or as it interacts with an optical component. Once these matrices are known, the field is propagated simply by multiplying a known field vector (usually at the laser), sequentially by the matrices representing each interaction it undergoes. This is the approach taken by the standard simulation package used to collect the dataset for this paper, FINESSE (Brown et al.).

In this process, the main computational cost comes from the fact that a) in order to capture sharper spatial features, higher order modes must be included, and the dimension of each scattering matrix grows in $\mathcal{O}(n^2)$, yielding a high dimensional system of equations to be solved, and b) computing the matrix elements themselves can be extremely computationally intensive, particularly when accounting for higher order effects like thermal lensing and scattering due to the finite mirror apertures.

Furthermore, in order to simulate a particular interferometer topology, multiple of these simulation subroutines must be run. This is because the interferometer must be "locked." In effect, the microscopic positions of the cavity mirrors must be adjusted in order to achieve resonance inside of the optical cavities, and determining this lock point requires iteratively adjusting mirror positions, re-running a simulation each time.

Deep learning based approaches have shown great success at accelerating computationally costly physics simulation routines such as this. Previous works have applied neural networks towards the design of compound lens systems (Yang et al. (2024)). Other works have shown that neural networks can directly solve partial differential equations similar to those that govern EM-field propagation (Li et al. (2020), Alkin et al. (2024)), or emulate hydrodynamic simulations of the cosmic web (Zhang et al. (2023)). Some of these approaches give the neural network direct access to the underlying physics, while in most cases, the approach is the data-driven one: large quantities of simulation data are given to the network, and the underlying physics is then learned. We note two distinctions between our problem setting and those covered by recent physics foundation models (Alkin et al. (2024), Herde et al. (2024)). First, we are not concerned with the time evolution of the field, only the steady state fields achieved inside the interferometer cavities. Secondly, the model does not need to emulate the entire field in between optics in the interferometer; the primary points of interest are in the interactions between the field and optics.

Graph neural networks in particular have been used simulating physical dynamics in settings where there is a natural spatial ordering, for example in mesh based simulation (Pfaff et al. (2021)) and in predicting the properties of molecules (Reiser et al. (2022)). GNNs can effectively capture local interactions between nodes (i.e. points on a mesh, or bonded atoms in a molecule), and through multiple rounds of message passing, can also incorporate long range dependencies. In the vein of this approach, we employ a graph-based architecture, that captures the spatial relationships between the different fields and optical components in the interferometer, and apply it to the problem of predicting steady state EM fields in interferometers.

# 4  GNNs FOR INTERFEROMETER SIMULATION

## 4.1  DATASET AND INTEFEROMETER MODEL

Because the EM-field is modelled as interacting with a sequence of optics, this lends itself very naturally to a graph representation. In this dataset, each mirror is split into four nodes, two for each side of the mirror, and with the incoming and outgoing fields treated as separate nodes. The edges in the graphs represent the spatial connection between fields. For example, the incoming field to a mirror will be connected to both the reflected field and the transmitted field, while the outgoing field from that mirror would be connected to the node representing the incoming field at the next optic

| Dataset | FP | Simple CC | Arm-SRC CC |
|---|---|---|---|
| Graphs | 30,000 | 5,000 | 30,000 |
| Nodes per Graph | 10 | 18 | 74 |

Table 1: Size of each dataset. Each graph has 3 node features, and each edge has 2 features.

it interacts with. Each node has two features, meant to represent the reflectivity and the radius of curvature of the optic.

At each node, the complex field amplitudes, beam parameter and powers of the even HG modes, up to sixth order, are recorded. We also provide helper functions to convert this information into the 2D spatial intensity distribution at each node.

This data is collected for three different interferometer topologies: a Fabry-Perot cavity, a coupled cavity, and a Arm-SRC coupled cavity setup, shown in Fig. 1. For each topology, a set of "base" configurations are chosen, and then a random walk is performed in the neighborhood of each of those base configurations, to collect the remaining data. The characteristics of each dataset are enumerated in Table

The dataset is sampled in the following way. We start at an "ideal" interferometer configuration (i.e. all cavities and the laser are perfectly mode-matched), and then stochastically perturb the interferometer parameters and run a FINESSE simulation at each step, which serves as our ground-truth data. For each interferometer setup, 30,000 samples are collected.

Over the course of the random walk, we modify the following optic properties: radius of curvature, reflectivity, and relative spacing. Other properties, like index of refraction are kept constant, as these are material properties, and perturbing them is not physically realistic.

### 4.2 MODELS

We train a network to predict the incoming and outgoing field power at each point in the interferometer, and a network to predict the spatial intensity distribution at each point in the interferometer, given the the interferometer topology, as a graph. Each node in the interferometer has three features: the radius of curvature of the wavefront, the reflectivity of the optic, and the angle at which it is oriented. Each edge in the graph has two features: its length and index of refraction. The model is trained on locked interferometer data but does not perform the locking procedure itself.

#### 4.2.1 LEARNING THE POWERS

Our task is to predict the incoming and outgoing powers at each optic in the interferometer. Points within optical cavities, particularly the arm cavities, have powers on the kilowatt (KW) scale, while powers exiting the interferometer may be on the milliwatt scale. To account for this scale separation, the power prediction model is trained to predict $\log P$, instead of the raw power.

The model architecture consists of 20 GATv2 layers (Brody et al. (2022)), followed by 6 feed-forward layers. LeakyReLU activation functions are used. Residual connections are placed between each pair of consecutive message passing layers, and between each pair of consecutive linear layers. For all the alternate model architectures tested, the number of layers of each type is kept the same. For example, the GraphTransformer (Shi et al. (2021)) architecture, we use 20 graph transformer layers, followed by the same 6 MLP layers.

The model is trained with a custom loss function, defined below:

$$\mathcal{L} = \frac{1}{n} \sum_n^N ||\mathbf{y}_n - \hat{\mathbf{y}}_n||_1 + \lambda ||\hat{\mathbf{y}}_n - \mathbf{A}^T \hat{\mathbf{y}}_n||_1 \tag{8}$$

where $\mathbf{y}$ is the ground truth power vector, $\hat{\mathbf{y}}$ is the model prediction, $\mathbf{A}$ denotes the adjacency matrix of the input graph, and $||x||_1$ denotes the L1 norm of the vector. The first term is the standard mean absolute error loss, and the second term is a regularization term that penalizes model outputs where the sum of incoming powers to a node does not equal the power at that node (i.e. conservation of energy).

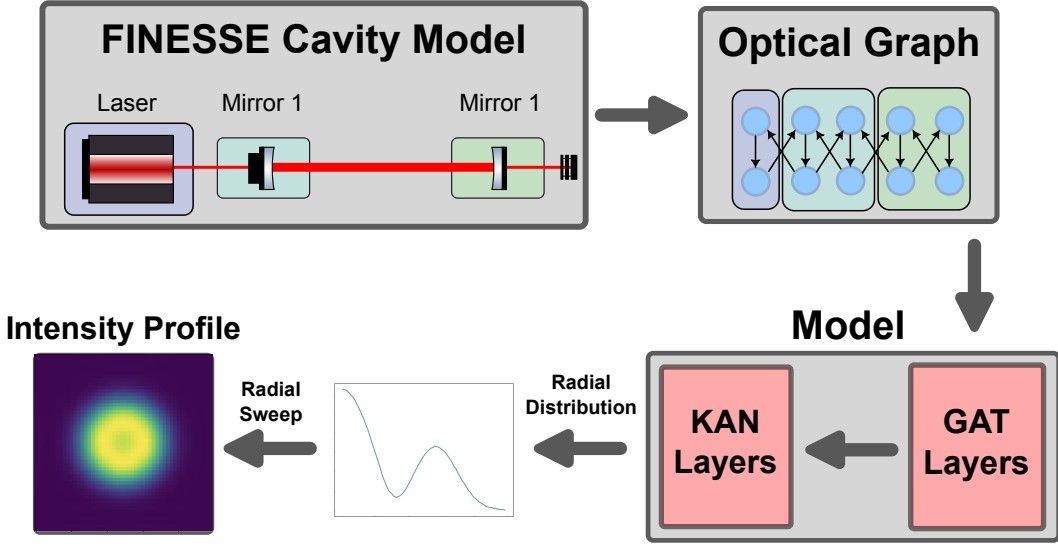

Figure 2: Interferometer simulation pipeline. The FINESSE interferometer model is converted into an optical graph, where each optic is broken down into a node for each incoming or outgoing field. This graph is fed to the model, which produces a radial intensity distribution, which is then rotated to produce the final intensity distribution.

We find that our model's performance is not very sensitive to the size or number of message passing layers, and increasing the depth of the network incurs a high computational cost.

### 4.2.2 LEARNING THE INTENSITY DISTRIBUTION

Limiting our model to predicting field powers does not capture the entirety of physically relevant phenomena for interferometer design. For example, we may be interested in the modal decomposition of the field, to ensure that power mainly stays in the TEM-00 mode. This kind of information is captured in the field intensity distribution. In this section, we describe the model architecture used to predict the intensity distribution.

The input interferometer graph is first passed through 15 GAT layers. The resulting node embedding is then passed to the Deep Kolmogorov Arnold Networks (KAN) described below.

Taking advantage of the radial symmetry of the field representation, as can be seen in Eq. 5, we pass the node embeddings to a DeepKAN, which learns to approximate the radial intensity distribution, which is then rotated to form the full 2D intensity profile. This enforces the physical requirement that in the cases we consider, the EM-field is azimuthally symmetric on the mirror surface. It also has the advantage of reducing the number of degrees of freedom for the intensity map from $\mathcal{O}(n^2)$ to $\mathcal{O}(n)$. This pipeline is shown in Fig. 2.

The choice to use Kolmogorov Arnold Networks is informed by their demonstrated proficiency in learning other special functions from physics, such as the spherical harmonics (Liu et al. (2024)) with fewer parameters than feed forward networks.

## 5 RESULTS

### 5.1 POWER PREDICTION RESULTS

For each interferometer topology, we report the mean absolute error (MAE) achieved by three different models in Table 2. We also show some sample correlation plots in Fig. 3.

The mixed model is trained on a training set comprised of 20,000 Fabry-Perot cavity simulations, and 4,000 Arm-SRC CC simulations. The remained two models are trained on 24,000 of a single simulation type.

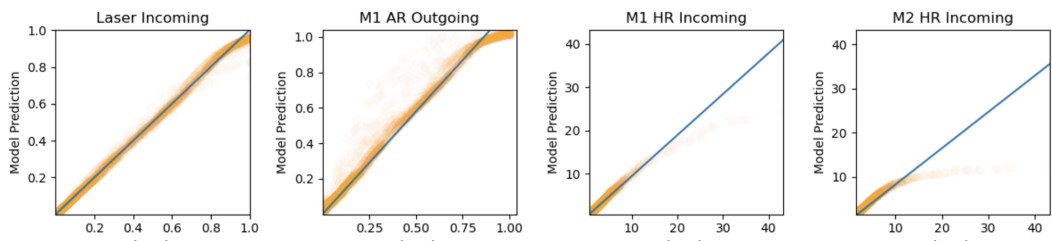

Figure 3: Correlations between dataset and mixed model predictions for the Fabry-Perot cavity. Units are in watts. Best fit lines are plotted in blue. A slope close to 1 indicates strong agreement between model predictions and ground truth data. In order, the slopes of the lines of best fit are $m = 1.00, 1.16, 0.95, 0.82$.

|  | Training Dataset | Arm-SRC CC | Fabry Perot | Simple CC |
|---|---|---|---|---|
| GAT + MLP | FP | $\infty$ | 0.52 | 2.94 |
|  | Mixed | 0.25 | 0.54 | 3.01 |
|  | Arm-SRC CC | 0.24 | 1.36 | 2.98 |
| GAT + KAN | FP | 0.58 | 0.57 | 0.89 |
|  | Mixed | 0.38 | 0.76 | 1.09 |
|  | Arm-SRC CC | 0.39 | 1.41 | 1.68 |
| MLP Only | FP | $\infty$ | 0.08 | 5857.71 |
|  | Mixed | 0.41 | 1.32 | 1380.19 |
|  | Arm-SRC CC | 4.89 | $\infty$ | 6.94 |
| KAN Only | FP | $\infty$ | 0.055 | 12.37 |
|  | Mixed | 0.19 | 33.98 | 38.91 |
|  | Arm-SRC CC | 0.29 | 4784.32 | 104.83 |
| GraphTransformer + MLP | FP | $\infty$ | 0.56 | 2.04 |
|  | Mixed | 0.34 | 0.65 | 1.71 |
|  | Arm-SRC CC | 0.39 | 1.34 | 1.65 |

Table 2: L1 Losses on test datasets for each interferometer topology. Results of our final architecture are summarized in the top section. The remaining rows compare this architecture to other combinations of GNN layers and MLPs/KANs.

The model trained purely on Fabry-Perot data generalizes very poorly to the more complex Arm-SRC CC setup. This is expected, given that the physical relationships between fields in the Arm-SRC CC setup are far more complex than those in the single cavity model. However, we note that even injecting relatively few training samples from the ALIGO dataset, as in the case of the mixed model, improves performance to the level of the model trained purely on ALIGO data.

We also compare our model against the following: an MLP and KAN, without the GNN, in which the input features and output features are concatenated into a single vector, to evaluate the importance of graph structure to the resulting predictions. We find that our GNN models consistently outperform a basic MLP and KAN, and in particular the MLP and KAN generalize extremely poorly to interferometer topologies not present in the training data, while the GNN models perform comparatively better.

Finally, we compare our model to an identical architecture with the GAT layers swapped for Graph Transformer layers. The GraphTransformer achieves comparable results, but is slower to train and run, and so we choose to use GAT.

A major concern in deep-learning based approaches to accelerating physics simulation goes as follows: Collecting training data is in and of itself extremely computationally intensive. Thus, if the model shows limited generalization capabilities, then its usefulness is highly limited. Here, we discuss the generalization power of these networks. We find that the models do show some generalization power, in that they are still relatively accurately able to predict powers in the coupled cavity topology, despite having never seen that in the training data. We also note that injection of relatively

few training samples of a more complex interferometer topology results in a model that is sufficiently accurately able to predict powers in it. However, improving this generalization to achieve equally low losses on the coupled cavity dataset as in the Fabry-Perot and Arm-SRC coupled cavity setups is an important step for future work.

### 5.1.1 Ablation of Power Prediction Model

| Test Dataset | Number of GAT Layers | | | | |
|---|---|---|---|---|---|
| | 1 | 3 | 8 | 15 | 20 |
| FP | 1.06 | 0.86 | 0.54 | 0.53 | 0.53 |
| Simple CC | 2.86 | 2.20 | 0.83 | 0.70 | 0.71 |
| Arm-SRC CC | 0.43 | 0.42 | 0.39 | 0.39 | 0.38 |

Table 3: Test losses as depth of model increases. In general, loss goes down, but with diminishing returns.

In Table 3, we illustrate the performance of our model as we vary the number of message passing layers included. In general, as depth increases the performance improves, but returns are diminishing. As we care about both performance and computational efficiency, we opt not to increase the depth further.

### 5.2 Intensity Prediction Results

We demonstrate the results of the intensity prediction model here. In particular, we show that the intensity prediction model is able to accurately predict varying spatial intensity distributions, as well as total field powers, in Fig. 4. The final row shows a typical failure mode of the GNN model with an MLP instead of a KAN, which fails to capture both the true intensity and the total power.

The model achieves an L1-loss of 27.2 $\frac{W}{m^2}$, while a model with an MLP instead of the KAN, with the same number of parameters, achieves an L1-loss of 58.4 $\frac{W}{m^2}$.

### 5.3 Computational Efficiency

The key objective is for our model to provide a heuristic estimate of interferometer physics, in order to accelerate large scale interferometer optimizations. To that end, we compare the inference time to simulations in two standard interferometer simulation packages, FINESSE and Stationary Interferometer Simulation (SIS) [Yamamoto (2007)] in Table 4.

We also provide run times for particle swarm optimization of a Fabry-Perot cavity using FINESSE simulations and using our model. The optimization objective is to maximize the resonant power in the cavity by mode matching the cavity to the laser with fixed beam parameter. The GNN based simulation and the FINESSE simulations both find the optimal solution, with 10W of circulating power in the cavity. We note that the reason why the full scale simulation does not see the expected 800x speedups the overhead of converting the FINESSE model into a format that the GNN takes. As this is not the main concern of this work, we did not put much time into optimizing this.

| | FINESSE Model | SIS Model | GNN (Power) | GNN (Intensity) |
|---|---|---|---|---|
| Single Simulation (s) | 2.857 | 14.932 | 0.018 | 0.011 |
| Fabry Perot Optimization (s) | 170.8 | - | 53.7 | - |

Table 4: Mean Simulation/Inference Times for a single run of a Arm-SRC CC-like topology, averaged over 100 runs, and a short, 100 step particle swarm optimization of the Fabry-Perot cavity. FINESSE simulations were collected with even modes up to 6th order, and finite aperture mirror maps. GNN inference times were collected on an NVIDIA A30 GPU.

### 5.4 Discussion

Interferometer simulation presents a particularly difficult problem for typical physics simulation approaches, due to the complex boundary conditions that the field is subject to. Unlike other domains,

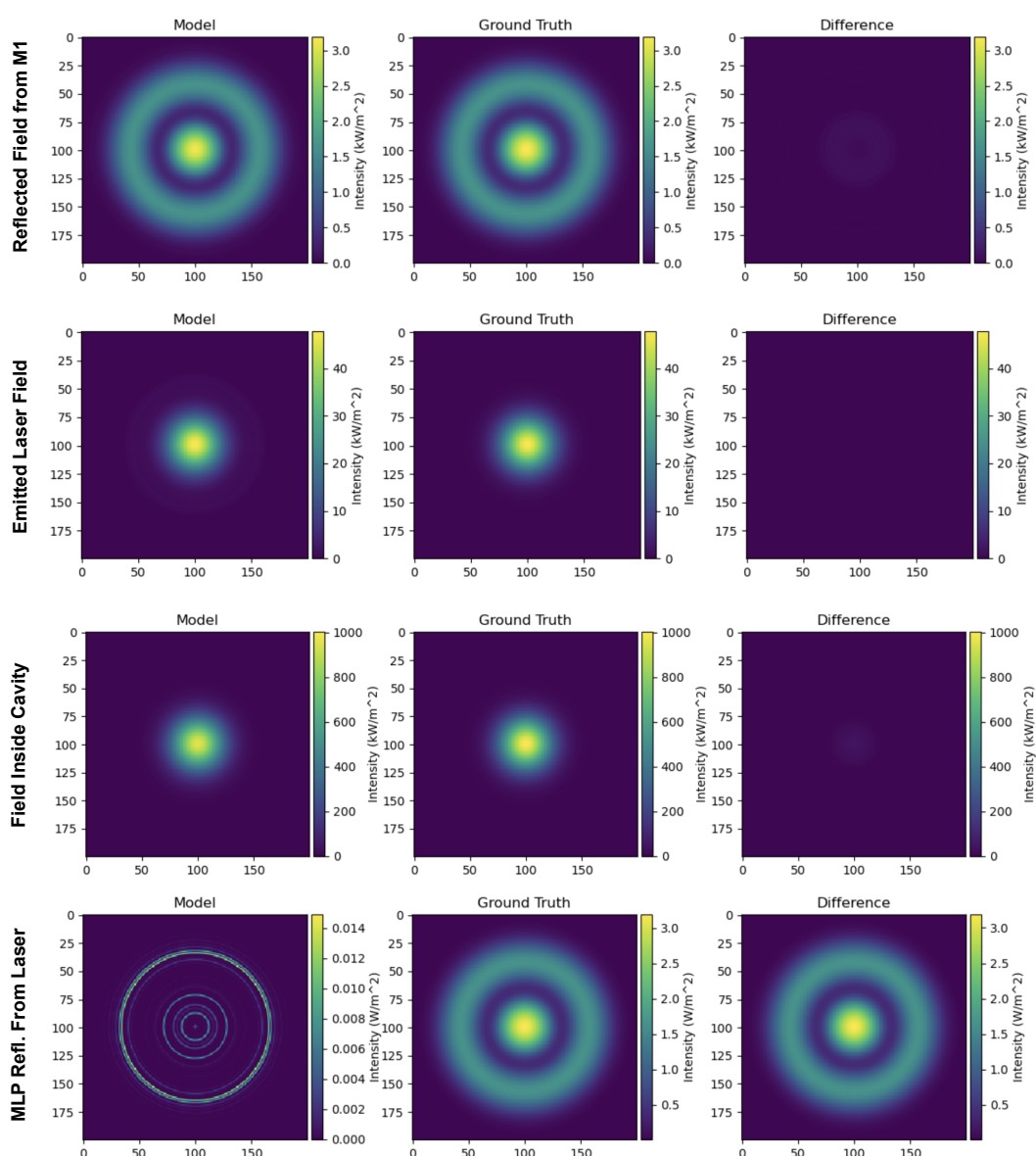

Figure 4: An example of intensity prediction results on coupled-cavity dataset. We note the following: The intensity prediction model accurately captures the differences in power at different points in the interferometer. Despite the similar, Gaussian profiles in the laser field and the cavity field, the intensity is almost an order of magnitude larger in the cavity, which the model captures. Equally importantly, the model captures the higher order modes present in fields at other points in the interferometer, namely in the reflected field, which contains a higher order modes, producing the second ring seen in the intensity profile. Errors in the first three rows are visible as a small bright patch in the center of the image; they are orders of magnitude smaller than the true intensity.

in which graph based ML approaches have been widely applied, such as structural mechanics or fluid dynamics, the quantities of interest here, like power, do not vary smoothly but instead jump sharply wherever the field interacts with an optic.

However, we also note that for our use case, emulated simulations do not need to be extremely accurate in order to be useful. Of course, the more accurate they are, the more useful they would be, but even a coarse approximation of the ground truth simulation admits significant speedups of optimization of future interferometer designs.

## 6 CONCLUSIONS

In this work, we introduce graph neural network models for simulating interferometer optical physics. We demonstrate that the models are capable of learning both basic quantities like the power inside the interferometer, and more complex characteristics, like spatial intensity distributions, while being hundreds of times faster per run than standard simulation packages. We note that our model shows some generalization capabilities, but the model does not achieve as low a loss on unseen optical topologies as on topologies in the training set. We see this as a key limitation of the model's viability in accelerating the design of future interferometers. In future works, we would like to explore ways of encoding the underlying physical quantities that would enable the model to understand the field propagation, and thus be more agnostic to the specific interferometer topology. For example, employing a mesh based approach similar to Pfaff et al. (2021) to learn the exact field amplitudes everywhere in the interferometer may generalize better.

In addition, in our work, we model a relatively small portion of the physics present in a full scale interferometer. The model we present is promising in that it already demonstrates the ability to accelerate low dimensional simulations, like those conducted in Richardson et al. (2022), but in order to be used for full scale interferometer design, it needs to be extended to incorporate more of the physics. In particular, in the future, we would like to extend our model to include higher order effects like point absorbers in the mirror surfaces, thermal lensing due to heating of the mirrors, and astigmatic beam shapes. These effects introduce new challenges in encoding feature representations for the 2D surface maps of each mirror.

In addition, a number of other works have applied generative models to similar problems in physics simulation. For example, Paganini et al. (2018) et. al apply a GAN-based framework to generate realistic particle showers to accelerate simulations of hadron calorimeters in the Large Hadron Collider. In future works, we could take this approach to model the distribution of a physical quantity of interest conditioned on realistic distribution of perturbations of a baseline topology.

In the larger context of instrument design, methods for accelerating simulations of interferometers would also open the gates to training reinforcement learning agents to more efficiently explore the parameter space of designs (Dworschak et al. (2022)).

Code for this paper can be found at: https://anonymous.4open.science/r/gnn-ifosim-36BE/

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
