# OpenReview forum: "Graph Neural Networks for Interferometer Simulations"
_ICLR.cc/2025/Conference — Submitted to ICLR 2025_

### Official Review · Reviewer_dynb · 2024-10-30

**Soundness:** 2
**Presentation:** 1
**Contribution:** 1
**Rating:** 3
**Confidence:** 3

**Summary:**

The authors present a GNNs based approach for accelerating simulations of optical interferometers. They demonstrate that their model can achieve significant speedups over traditional simulation methods while maintaining reasonable accuracy. However, the paper has several notable limitations.

**Strengths:**

1. The paper proposes a GNNs based method that can simulate electromagnetic (EM) field propagation in optical cavities at a fraction of the computational cost of traditional methods.
2. The authors provide a dataset on three interferometer topologies, which may offer potential future use in benchmarking for similar studies.

**Weaknesses:**

1. Insufficient Novelty in Model Design. The model primarily relies on standard GAT layers and KANs without substantial adjustments specific to interferometer physics.
2. Incomplete Comparisons with Existing Methods. The results reported are insufficient for an in-depth assessment. Without comparisons with other methods, the use of GNNs feels unjustified, and the work seems as an attempt to apply GNNs to a novel domain without fully validating its effectiveness.
3. Poor clarity and presentation. The content and figures do not meet high standards and lack clarity. For instance, while the authors emphasize the importance of instrumentation design in the abstract and introduction, the paper has minimal discussion on actual instrumentation design.

**Questions:**

Please see Weaknesses.

---

> ### Author Response · Authors · 2024-11-15
>
> We thank the reviewer for the feedback provided. However, we would like to clarify a few points.
>
> 1. The work does for the most part use standard layers in our model. However:
>
> a. We do modify some aspects of the training procedure to reflect the physics of our system. For example, we enforce energy conservation in the power prediction model by using a modified loss function, and we use the symmetries of the EM field to reduce the number of parameters of the intensity prediction model. While not direct improvements to GNN architecture itself, we feel that domain specific considerations like these make our work an example of techniques that can be used to enforce physical constraints in ML for science applications.
>
> b. The goal of our work is not to provide architectural improvements to a network architecture, but rather to demonstrate an application of GNNs. While future work could certainly pursue larger architectural changes (for example, drawing from research in neural operator learning or multiphysics emulation), this work is a first foray into applying ML to interferometer simulation (and more broadly scientific instrument design).
>
> c. We will also point out that the paper explores certain relatively new approaches in machine learning for science, like Kolmogorov Arnold networks, which were only introduced earlier this year, and are as of now still being studied to determine if and where they provide advantages over MLPs. Our work clearly demonstrates a use case in which KANs greatly outperform a similar MLP based architecture, and thus we feel that this paper may also be of interest to the broader ML community.
>
> 2. We agree with the reviewer’s criticism that the comparison to other model architectures is insufficient. We note that this is the first work applying ML to interferometer simulations, and thus we cannot compare it to other work done for this specific problem. However, we will add more detailed studies comparing our model performance to graph based and non-graph based ML model architectures, in order to help clarify the design decisions that we made.
>
> 3. We understand that the certain aspects of the content or the figures may have been unclear. If the reviewer has specific questions or critiques, the authors would be happy to hear them and to clarify them both here and in the paper itself. On the specific point of instrument design, we discuss this in the introduction and the conclusion, as this is the context in which this work was conceived. However, the specific challenges of instrument design more broadly are less relevant to this work than the specifics of interferometer physics that we discuss.

---

### Official Review · Reviewer_Ltfq · 2024-10-31

**Soundness:** 2
**Presentation:** 3
**Contribution:** 2
**Rating:** 3
**Confidence:** 3

**Summary:**

This paper applies graph neural networks (GNNs) to interferometer simulations, specifically modeling LIGO’s optical physics. The authors show that GNNs simulate electromagnetic field propagation accurately while running 815 times faster than traditional methods, which is supposed to enable efficient optimization of interferometer design. They also release a dataset of high-fidelity simulations for benchmarking future models in this domain.

**Strengths:**

* The paper introduces a new dataset that can be used to predict interferometer simulations.
* A GNN model that includes a KAN layer is introduced for the new data.
* The model is empirically evaluated and shows some promising initial results.

**Weaknesses:**

* Incomplete results: I would expect that the authors compare the results of both the GNN and MLP trained and tested on all possible combinations of datasets. However, Table 1 only contains a subset of these results. In particular, results for the MLP are only provided when trained on the mixed dataset. This makes it impossible to deduce a fair comparison from these results. Also, these partial numbers are only provided for power prediction, for intensity prediction only two numbers are mentioned in the text. Suggestion: Provide a full table showing results for both GNN and MLP models trained and tested on each individual dataset (Fabry-Perot, Half ALIGO, Coupled Cavity) as well as the mixed dataset. Additionally, quantitative results for the intensity prediction task can also be included in the table.
* The GNN is only compared to an MLP, which makes the analysis quite limited. Other methods should be tested, for example a KAN without the GNN. This would highlight the contribution that the GNN makes. As is, it is impossible to say whether the KAN alone might be the main contributor to better performance or not. Further, the GNN should also be tested with an MLP as a drop in replacement for the KAN to verify that it works better (as was partially done in the intensity prediction, but just for one setting). Suggestion: Include an ablation study that compares:
    * The full GNN+KAN model
    * A KAN-only model without the GNN component
    * A GNN with an MLP replacing the KAN

This would help isolate the contributions of each component. Ideally, these comparisons would be done across all datasets and tasks (power and intensity prediction) for consistency.

* Essential information on the setup and parameter ranges that were used to generate the ground truth data are missing in the paper (for example the parameter ranges that were used and how these were sampled).
* The GNN architecture and constructed graphs are very simple and no advanced methods were tested. More recent GNN variants like GraphTransformers or exploring edge feature incorporation methods would strengthen the paper in that regard.

**Questions:**

* This is a suggestion: Add an ablation study for the KAN and different GNN layers (not just GAT).
* Figure 3 is never referenced in the text
* I’m not an expert in interferometer spectography, so I cannot quite judge the impact of this work. Is this something that could be deployed as is or are the datasets too simplistic? Could you maybe provide a more detailed discussion on the practical applicability of your current model and dataset? What are the limitations of the current approach and what additional steps would be needed to make the model deployable in real-world interferometer design scenarios?
* It is unclear for me whether the direct comparison of a GNN with Finesse in terms of runtimes is really adequate. Is Finesse also just computing the same values that the GNN is computing here?

Overall, the experimental evaluation, which is a crucial part of this paper, is too limited and needs to be reworked before this paper should be accepted. I’m happy to reconsider my rating once the full results are provided.

---

> ### Author Response · Authors · 2024-11-15
>
> We thank the reviewer for their detailed review. We agree with the reviewer that a more detailed ablation study and more complete comparisons of results would improve the paper; they shall be added before the end of the discussion period.
>
> In response to the reviewer’s questions and comments:
>
> 1. We will add the ablation study. We would like to clarify that some of the more advanced architectures were in fact tried. In particular, we did experiment with Graph Transformers, but saw no significant advantage; this will be included in the ablation study.
>
> 2. Edge features are used; in particular, the separation between optics and their indices of refraction are encoded as edge features in our graph.
>
> 3. Figure 3 demonstrates the success of the model at the power prediction task. We will add discussion of this to the paper.
>
> 4. Regarding the question on FINESSE: In order to compute the power at each node, FINESSE does need to compute other quantities (in particular the amplitudes of each individual Hermite Gauss mode, and the beam parameters of each cavity). However, we still feel that this is a fair comparison, for two reasons. First, we consider the power to be a valuable metric for measuring the quality of a particular interferometer design. The fact that FINESSE must compute other quantities in order to get to that power is precisely where the computational cost of the FINESSE simulation arises from, and why we believe that this model serves a useful function. Secondly, the intensity prediction model predicts the spatial distribution, and thus indirectly predicts the modal decomposition, which is also what FINESSE computes under the hood. Seeing as the intensity prediction model achieves similar speedups to FINESSE/SIS, we feel that this comparison is still fair.
>
> 5. Regarding the direct applicability of this model to gravitational wave interferometry, we will add more discussion of this in the revision. However, as a short synopsis: This model is promising in that it does capture many of the important physical phenomena present in the interferometer. Namely, it is able to predict spatial intensity profiles, and thus modal decompositions of the EM-field. This is enough to be useful in accelerating optimizations of basic design parameters like cavity lengths and mirror radii of curvature. The interferometer topologies considered here range from simple (Fabry-Perot) to close to full interferometer complexity (half-ALIGO). Therefore, we feel that the most important limitation of this work is not in the complexity of the interferometer topologies considered, but in the fact that we only model a subset of the physics (power and spatial intensity distributions). To increase the applicability of these models, we would ideally need to account for more of the physical effects that are present, such as higher order mode scattering due to thermal lensing and surface defects in the mirrors. However, we will reiterate that the goal of this work was not to model all of the complexity of the LIGO interferometer, but instead to demonstrate that GNNs are capable of capturing the complex dependencies of the field propagation in interferometers, and thus are a promising candidate for simulation acceleration.

---

> ### Author Response · Authors · 2024-11-28
> **Summary of Revisions**
>
> We thank the reviewer again for the comments. We hope the revisions we made have satisfactorily addressed them.
>
> 1. We would like to clarify that for the intensity prediction model, the model described as “MLP model” still included GAT layers. We meant that an MLP was used to learn the radial intensity distribution instead of a KAN. This has been clarified in the text.
> 2. We have added a much more detailed and full presentation of results in Table 2.
> 3. We have added more discussion on the next steps required to make the model more useable in interferometer design.

---

> > ### Comment · Reviewer_Ltfq · 2024-11-30
> >
> > Thanks for the clarification. I looked at the new results you provided and still have a few questions:
> > * What about the full results for the intensity prediction task?
> > * What about the parameters used to generate the ground truth data?
> > * The GraphTransformer+MLP seems to be on par with GAT+KAN in many cases. Since the KAN is an improvement over the version with the MLP, is there a reason why you didn't test it with the KAN?
> > * Did you only run the experiments with a single seed or why are no standard deviations or similar reported?
> > * How exactly was the graph transformer implemented? Did you use any positional encodings? Did you only test with 20 graph transformer layers (this seems a bit excessive)?

---

> ### Author Response · Authors · 2024-11-30
>
> 1. Due to limited time and compute resources, we had to choose which experiments to prioritize. The intensity prediction model takes significantly longer to train, and so we chose to prioritize the power prediction task, as this is the more important metric for interferometer design. However, we can prepare more complete results here for the camera-ready, should the paper be accepted. In particular, we can do a similar comparison in the style of Table 2 for the intensity prediction task.
>
> 2. The authors are unclear on what the reviewer means by "parameters". The ground truth data is generated in the following way: A "base" interferometer design is randomly chosen, and then a random walk in parameter space is performed starting at the base design. This is repeated for a large number of base designs. If the reviewer could clarify which parameters they are referring to, we would be happy to clarify what values were used.
>
> 3. We agree that the GraphTransformer is roughly on par with the GAT+MLP combo. However, we chose not to evaluate it much further, again due to the time constraint, and because the inference time using GraphTransformer was higher than with GAT, without performance gain. As the core objective of our model is to enable fast simulation, we felt that it was lower priority to attempt to explore models that provide marginal improvements in accuracy at the cost of higher computational cost. A similar logic applies to the GAT+KAN model. KAN is roughly *an order of magnitude slower* at inference time than the GAT+MLP combo, and additionally performs slightly worse on two of the three tested topologies. We would be happy to add exact figures for the computational efficiency of each tested model.
>
> 4. For the experiments added in the revision, we tested with only a single seed, yes. This seed was fixed across all experiments.
>
> 5. The graph transformer implementation used is the one implemented in PyTorch Geometric (https://pytorch-geometric.readthedocs.io/en/latest/generated/torch_geometric.nn.conv.TransformerConv.html). No positional encodings are used, and yes, we used the same 20 layer architecture as the GAT.

---

> > ### Comment · Reviewer_Ltfq · 2024-12-02
> >
> > Regarding the parameters: Could you clarify the exact setup for the "base" interferometer design, particularly the distances between objects and other relevant parameters to generate the data? When the random walk is performed in the parameter space, what specific parameters are being adjusted, and are they sampled uniformly? My concern is that the data generation process should be sufficiently clear and reproducible, allowing for future extensions and a better overall understanding.
> >
> > > KAN is roughly an order of magnitude slower at inference time than the GAT+MLP combo, and additionally performs slightly worse on two of the three tested topologies.
> >
> > This statement is a bit unclear to me. Isn't the main objective of your paper to use GAT+KAN?
> >
> > In general, I feel that the results still have significant gaps, and the experimentation is lacking in several areas. Some of the choices made appear somewhat ad hoc, which makes it difficult to fully evaluate the paper and give it an accepting score.

---

> > > ### Author Response · Authors · 2024-12-03
> > >
> > > Regarding the parameters, there are multiple base parameters that were used. As part of our code release, we can include a file that spells out these base parameters that are used. In addition, the actual datasets are released as part of our code release. The parameters that are adjusted are listed in the dataset generation portion of the paper. They are the radius of curvature of the optics, the relative spacing, and reflection coefficients of the optics. The adjustments are sampled uniformly.
> > >
> > > As for the question regarding KAN, the results we present demonstrate that the GAT + MLP combination is sufficiently good for the power prediction task, but insufficient for the intensity distribution prediction. Our objective is primarily to show that the interferometer emulation task is possible with GNNs, though the specific architecture may vary depending on the task.
> > >
> > > We hope this clarifies some of the reviewers concerns.

---

### Official Review · Reviewer_f3YH · 2024-11-03

**Soundness:** 3
**Presentation:** 3
**Contribution:** 3
**Rating:** 6
**Confidence:** 3

**Summary:**

This paper applies GNN and KAN to simulate LIGO instrumentation models and shows that it can accurately capture the optical physics at play while achieving 815x faster than SOTA EM numerical simulation packages. It also provides a high-fidelity dataset of optical physics simulation for three interferometer topologies for benchmarking.

**Strengths:**

+ Shows the necessity of using GNN rather than MLP for feature extraction and generalization, as well as the benefit of KAN, as compared to MLP in more actually addressing spatial features such as the prediction of varying spatial intensity distribution.
+ Results show significant speedups than numerical simulation

**Weaknesses:**

Authors should provide more details & clearer explanation about their datasets and the mapping from optics of EM field to a graph. I would highly recommend showing a figure to illustrate an example, including the node features and edge features.

About the dataset, can you provide more characteristics? e.g., how many data points, how many nodes, etc. in a table. Also, why the full ALIGO is not covered in the dataset, which seems to be the practical establishment for LIGO.

I would expect more concrete discussion on the advantages of using KAN than MLP for such a type of physics-information models.

**Questions:**

What are the practical benefits of using this AI model in addition to reducing optical simulation time and what are their implications? Have you observed that the AI model can provide better manufacturing design parameters that are more robust?

Can you provide more insights on the rationale of designing such a 20-layer GAT + 6 layer MLP or 1 layer KAN kind of model? It seems to me the lack of ability to generalize for MLP is due to overfitting by using the 6-layer MLP.

---

> ### Author Response · Authors · 2024-11-15
>
> We appreciate the reviewer’s time in providing this review and thank the reviewer for the useful feedback.
>
> Regarding the reviewers questions:
> “What are the practical benefits of using this AI model… Have you observed that the AI model can provide better manufacturing design parameters?”
> Other works (Richardson et. al 2022, for example) identify the existence of better manufacturing parameters than the current LIGO design, but have been restricted to exploring a subset of the interferometer’s degrees of freedom, due to the computational cost of simulation. Thus, the main benefit of our AI model, as the reviewer states, is in the accelerated simulation times. During the design phase of GW detectors, there is a vast parameter space that must be searched to identify the design which will maximize the sensitivity. This design phase is currently ongoing for Cosmic Explorer, a future interferometer, and so tools to improve this optimization/exploration of design space are intended to assist in that effort. This will be clarified in the paper text.
>
> “Can you provide more insights on the rationale of designing such a 20-layer GAT + 6 layer MLP or 1 layer KAN kind of model?”
> This was an empirical decision, based on experiments conducted with different architectural parameters. A more detailed ablation study will be added to the paper, along with a discussion of the architectural choices.
>
> Regarding the comments on clarifications to the dataset:
> We will modify the figures and explanations to clarify how the dataset is mapped to a graph, as well as the requested table.
> For the reviewer’s ease, we will provide a short description here as well. The datasets each contain around 10,000 graphs. Each graph represents one interferometer simulation, where each node in the graph represents a component of the EM field at a particular optic (mirror, laser, etc.). The edges represent the properties of the space between those optics (separation distance, index of refraction, etc.). A full-ALIGO dataset is not provided, as while it would be the end goal of such a study to model a full-ALIGO like interferometer, the half-ALIGO model captures much of the important physics also present in the full-ALIGO case.

---

> ### Author Response · Authors · 2024-11-28
> **Summary of Revisions**
>
> We thank the reviewer again for the comments. We hope the revisions we made have satisfactorily addressed them.
>
> 1. We add a summary of datasets in Table 1, along with more description of how the data was collected in the text of the paper (Section 4.1).
> 2. We add a table comparing the performance of the network with depth, to explain why we chose the architectural parameters that we did.

---

### Official Review · Reviewer_X5yy · 2024-11-05

**Soundness:** 4
**Presentation:** 4
**Contribution:** 3
**Rating:** 5
**Confidence:** 4

**Summary:**

The authors present a graph neural network purpose-constructed for interferometer simulations, which combines a graph attention network with the recently introduced Kolmogorov-Arnold networks. Deriving networking constraints from the physical symmetries belying the simulation approach to interferometer simulations, the two are combined to abide by the physical symmetries of the problem. The proposed model is evaluated on 4 successively more difficult interferometer topologies, and evaluated for its predictive capability as well as computational efficiency.

**Strengths:**

The paper is very well-written, and all network design decisions are motivated out of the demands by the physical problem which are introduced in the requisite detailed manner. Especially the description of the physical problem is very well-written and enables an easy understanding of the imposed demands.

**Weaknesses:**

The weaknesses of the paper can on a high level be summarized with lack of depth of the evaluation, a lack of embedding into the wider literature, and imprecision in a number of key claims.

**Lack of depth of Evaluation**
- Specifically table 1 seems to only capture a limited window of the design space. The evaluated models, as well as the dataset evaluations could be improved considerably with a limited amount of effort, such as on the architectural side evaluate "GAT + KAN", "GAT only", "KAN only", and "MLP". Each of these 4 would then be evaluated with the 3 dataset specifications "FP Only", "Mixed Dataset", and "Half ALIGO Only". In addition the current caption is imprecise in its description of what is happening in the present table.
- Equivariant layers for GNNs being readily available, it is unclear to the reviewer while the GAT + KAN hybrid is not contrasted to a pure GNN with Equivariant layers motivated out of the physical symmetries of the interferometry problem as a point of comparison, and a further point of comparison for the computational efficiency of the introduced architecture.
- The exact details of the GNN architecture are not further specified before the results section, while it is specified that it used 15 GAT layers + deep KAN, but for the full reproducibility of the results it would be beneficial to provide a succinct table containing the details of the overall architecture plus the design specification of other models used in the evaluation.
- With regards to the evaluation of the computation efficiency, please see the imprecision in key claims below.

**Lack of Embedding into wider Literature**
- While the work on GNNs for fluid simulations is mentioned throughout the paper, the paper yet contradicts literature in claiming for these GNNs to not be applicable to the interferometry problem. Shock simulations, which exhibit the sharp jumps characteristic to interferometry are an integral part of fluid simulations, and works such as Poseidon, as well as the widely used PDE-Bench contain shock simulations, and are hence able to represent those sharp jumps. Setting the presented work in relation with GNNs for shock simulations, and potentially even testing one of the fluids-trained models on interferometry such as Poseidon by fine-tuning it on the introduced dataset would benefit the paper greatly.
- The training across multiple interferometer configurations bears close resemblance to the Multiple Physics Pretraining introduced in _Multiple Physics Pretraining for Physical Surrogate Models_ by McCabe et al., I would urge the authors to relate their work to the MPT approach, and if applicable in the author's eyes consider other recent Transformer or GNN architectures for PDEs trained across multiple problem settings such as the models trained on PDE-Arena, or PDE-Bench, as well as large models like the Universal Physical Transformer of Alkin et al., and the Poseidon model series by Herde et al. The latter is furthermore trained on shocks, and as such should be able to be fine-tuned for the interferometer simulation as it is natively trained to be able to handle the sharp jumps of interferometry (line 431).

**Imprecision in key claims:**
- The authors claim an 815x speedup, while at the same time emphasizing the benefits of outputs with a lower fidelity. At the same time, the abstract leaves the impression that this speedup is realized at the same simulation fidelity. It is furthermore unclear to the reviewer if those fidelities are actually the same. Going from the results table, this does not seem to be the case. In addition the reviewer asks for further clarification if both are performed on the same platform. The GNN network does seem to be running on a GPU, but it remains unclear if the FINESSE model is also running on a GPU.

**Questions:**

Line 258-260: Ablations of model design decisions, why don't the authors consider using their spare space to extend the paper with an in-depth ablation study? Going from the mentioned paragraph, you have done these ablations, and mention them, but have not synthesized the ablations into a dedicated ablation section, which could improve the paper.

What is the GNN 815x faster than, and how does the fidelity compare? A way to seek to quantify this relation would either be a discrete fidelity on the x-axis, with the y-axis representing the speedup, or the fidelity could also be a continuous axis, where one would use the Pearson correlation [1] between the full fidelity result of the FINESSE simulation, and the output of the GNN. Further depth could be added to this plot by performing the same calculations for a GAT-only architecture, a KAN-only architecture, and the MLP used in the main results section.

[1] https://mathworld.wolfram.com/StatisticalCorrelation.html

---

> ### Author Response · Authors · 2024-11-15
>
> We thank the reviewer for the in-depth and constructive critique. At a high level, we agree that the paper would benefit from a more in-depth ablation study and better embedding into the existing literature. As doing these requires running further experiments and understanding the references that the reviewer provides, we will require some time to revise the paper to address these critiques. We will do so by the end of the discussion period.
>
> Regarding the questions that the reviewer poses about the simulation speedups and fidelities, we provide the following clarifications:
>
> We believe that part of the confusion is the dual contexts in which we use the word “fidelity.” When we refer to costly, high-fidelity optical simulations, we mean that in order for the optical simulations to produce physically accurate results, a large number of higher order beam modes must be included. Our GNN model operates at this high fidelity. When we say that the model predictions would still be useful at low fidelities, we mean that a somewhat inaccurate model is still useful as a heuristic for search space exploration. We have changed the terminology to say that the model predictions do not have to be very high accuracy.
>
> The GNN is 815x faster than an SIS simulation of the Half-ALIGO interferometer topology. The GNN is 159x faster than a FINESSE simulation of the same topology. The FINESSE model is run on CPU, as FINESSE does not currently have GPU support. The SIS simulations are also run on CPUs, though SIS does support GPUs. We will include a comparison with SIS run on GPU as another baseline. All three models are run at equivalent “fidelities”.
>
> We will add revisions regarding the depth of evaluation critique once the experiments conclude.

---

> > ### Author Response · Authors · 2024-11-28
> > **Summary of Revisions**
> >
> > We have added the following changes to address the your comments.
> >
> > 1. In Section 3, we add some discussion about recent works like the Universal Physics Transformer and Poseidon models suggested by the reviewer. We draw the following distinctions between their approach and ours: a) We are not interested in time evolution, but only the steady state fields and b) we are not interested in the full field, but only its modal decomposition at the optic surfaces. The free field propagation between optics is a well understood phenomenon, and is fairly cheap to simulate, unlike the fluids applications being handled by the Poseidon model. These two criteria make those models somewhat inapplicable for our task without some modification.
> >
> > 2. We include a much more detailed ablation study and comparison between model architectures in Table 2.

---

### Meta-Review · Area_Chair_wc6T · 2024-12-20

**Metareview:**

The paper proposes to adapt GNNs to model optical physics in Laser Interferometer Gravitational-wave Observatory (LIGO), using a new KAN layer specific to the optical simulation data. The work also introduces a dataset of high-fidelity optical physics simulations for three interferometer topologies, which can be used for future benchmarking. It is the first time that GNNs were applied to model optical physics. However, the reviewers point out incomplete evaluations between variants of the GNN, and questioned the claim about the computational efficiency. Therefore, the paper is rejected.

**Additional Comments On Reviewer Discussion:**

The reviewers point out that the results have significant gaps, namely the paper lacks comparisons between GNN and MLP trained and tested on possible combinations of datasets, as well as conducting ablations on the usefulness of the new KAN layer. The reviewers also noted that one of the major claim about 815x times faster runtime comes at a cost of reduced accuracy. The authors did not sufficiently clarify these claims during the rebuttal. The paper does not meet the high standards of ICLR conference.

---

### Decision · Program_Chairs · 2025-01-22

Reject